# Sapling: Successive Adaptation and Compression with Layer Dropping for LLMs

## Abstract

Specializing Large language models (LLMs) for local deployment and domain-specific use can deliver state-of-the-art performance while meeting latency and privacy requirements. However, conventional task-specific adaptation does not show both memory saving and inference speedup at deployment time. Practical compression techniques like quantization and pruning require hardware support or system optimization to achieve measured inference speedup. We propose Sapling, which can retain LLMs' capacity in a specific knowledge domain and achieve inference speedup on any hardware and deep learning systems by reducing the model depth. Sapling is based on the knowledge localization phenomenon we empirically observed and verified on LLMs, and achieves model compression via successive layer dropping. We evaluated Sapling on LLaMA-7B. At inference time, the models adapted on medical, legal, and financial datasets have all demonstrated reliable performance, comparable memory saving, $1.2$ to $8.5\times$ inference speedup on consumer-level hardware compared to state-of-the-art quantization algorithms, depending on how well the algorithms are supported by efficient accelerator kernels.

## 1 Introduction

Large language models (LLMs) are gaining prominence, with a growing interest in specializing them for specific domains like medicine (Thirunavukarasu et al., 2023), law (Yue et al., 2023), and finance (Wu et al., 2023b), and deploying locally to address latency and privacy concerns in sensitive data use cases. For example, understaffed clinics can benefit from deploying medical-specialized LLM-based chatbots on local devices. However, the sheer amount of memory and computation required for inference present significant barriers to deploying specialized LLMs in such resource-limited scenarios.

Post-training quantization (PTQ) is a primary technique to fit LLMs into resource-limited environments for inference, by reducing the bit precision of LLMs' weights to as low as 4 or even 3 bits, without significantly degrading model performance. However, to translate theoretical inference speedup into wall-clock speedup, most PTQ methods (Dettmers et al., 2022; Xiao et al., 2023; Frantar et al., 2022; Lin et al., 2023) require efficient kernels and even additional support from hardware vendors to provide corresponding quantized computational operators, which, unfortunately, is not easily accessible. Consequently, incorporating the latest quantization techniques in practice often *slows down* model inference, evidenced in Table 1, with the exception of AWQ (Lin et al., 2023), which is equipped with a decoding implementation that supports quantized weights. Similar results were observed on many post-training LLM pruning algorithms such as Kwon et al. (2022); Frantar & Alistarh (2023a); Sun et al. (2023) which require hardware support for unstructured and structured sparse tensor operations.

In light of these limitations, this paper explores a new way of compressing LLMs. We are motivated by recent findings about *knowledge localization* (Meng et al., 2022b; Li et al., 2023) in LLMs. Particularly, knowledge localization shows that middle layers in LLMs contribute more to the domain-specific knowledge generation process (Meng et al., 2022a; Azaria & Mitchell, 2023). Within each layer, attention modules are more likely to extract general semantic correlation while MLP layers are more task-specific (Geva et al., 2020). Inspired by this phenomenon, we hypothesize that each decoder block, especially its MLP layer, weighs differently for different knowledge domains. By

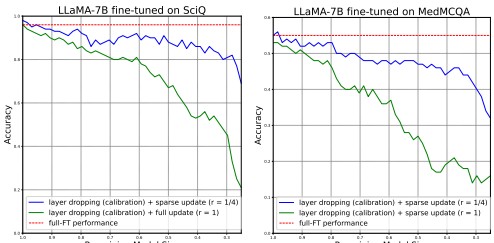
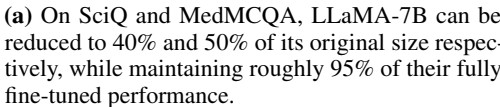
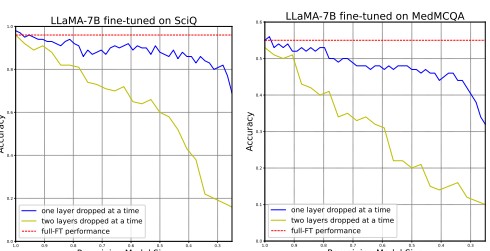

**(a)** On SciQ and MedMCQA, LLaMA-7B can be reduced to 40% and 50% of its original size respectively, while maintaining roughly 95% of their fully fine-tuned performance.

**(b)** Sapling's performance is upper bound by the granularity of dropping 1 layer at a time. Dropping 2 layers at a time in general performs worse. The results generalize to other tasks.

**Figure 1.** We ran two successive layer-dropping experiments on LLaMA-7B. One is performed on a common-sense QA benchmark, SciQ, and the other on an medical QA benchmark, MedMCQA. Performances are evaluated on a subset of the validation set.

dropping less important layers during fine-tuning, we aim to achieve a balance between memory footprint, inference speed, and domain-specific performance with a shallower specialized LLM.

To validate this hypothesis, we conducted extensive layer-dropping experiments on domain-specific datasets (Pal et al., 2022; Chalkidis et al., 2021; Maia et al., 2023), in which we drop one insignificant layer after one epoch of fine-tuning. Layer-dropping results, as shown in Figure 1a, indicate that up to 60% of the parameters can be dropped without significant performance degradation. On the other hand, model specialized to one domain via layer dropping show significantly compromised performance on a different domain. This verifies our hypothesis that different layers of a pre-trained LLM store different domain knowledge.

Building on these findings, we introduce *Sapling*, a model compression framework employing successive layer dropping, capable of compressing LLMs to $> 50\%$ of their original size while preserving their domain-specific performance. Sapling uses a calibration dataset to identify and drop the most insignificant layers after each iteration. We also developed a sparse update scheme to only train on the most important layers while neglecting the ones that might eventually be dropped.

LLMs pruned via Sapling show comparable ML performance on domain tasks compared to the fine-tuned full model, with far fewer parameters – hence significantly decreased memory and flops requirement at inference. Unlike PTQ or existing pruning methods, Sapling does not alter precision nor introduce sparse computation, therefore it does not depend on specialized kernels. Since Sapling is performed during fine-tuning, it is orthogonal from other model compression techniques.

The key contributions of this paper are: (1) We observe and empirically verify the layer-wise *knowledge localization* phenomenon on contemporary LLMs; (2) We design Sapling, a new approach for model compression. Sapling prunes LLMs during fine-tuning, by discovering and removing unimportant layers. (3) We show Sapling achieves $> 2\times$ memory saving and $> 2\times$ inference speedup in comparison with the model in full size on medical, legal, and financial domain-specific datasets. We also show Sapling's ability to realize $1.2 - 8.5\times$ inference speedup than the baseline quantization and pruning approaches. As a side benefit, Sapling offers a flexible "continuum" of target model sizes compared to other compression methods.

## 2 RELATED WORK

**Task-specific adaptation.** A typical workflow for task-specific adaptation is to first fine-tune (Wu et al., 2023a; Yang et al., 2023; Huang et al., 2023b;a) or even pre-train (Wu et al., 2023b; Cui et al., 2023; Shah et al., 2023) LLMs on task-specific datasets before applying any of the following three model compression techniques for reliable performance during inference: quantization, distillation, and pruning. In our case, we adopt layer-dropping to compress the model step-by-step *during* fine-tuning, i.e., we adapt LLMs to domain-specific tasks by identifying and retaining important layers for the target domain.

**Quantization** effectively mitigates memory consumption by reducing the bit-widths of LLMs' weights and activations. Quantization has featured its ability to retain LLM's zero-shot ability

**Table 1.** Deployment-time model inference overhead breakdown (LLaMA-7B, on single V100 GPU, sequence length 512 , batch size 1). The Overhead entry refers to the overhead of running the corresponding model compression algorithm after fine-tuning. The Final Mem entry refers to the ratio of final compressed model size versus the original model size in memory.

| Techniques | Overhead (s) | Inference Throughput (tokens/s) | Final Mem |
|---|---|---|---|
| FP16 | N/A | 16.6 | 100% |
| LLM.int8() | 57.3 | 4.1 | $\geq 50\%$ |
| GPTQ-int4 | 371.5 | 7.2 | $> 25\%$ |
| AWQ-int4 | 542.9 | 29.3 | $> 25\%$ |
| Sparse-GPT (2:4) | 215.4 | 21.2 | 100% |
| Masked Pruning | 253.2 | 17.7 | 100% |
| Activation-based Pruning | 0.54 | 16.1 | 100% |
| Sapling (40%) | N/A | **34.9** | $\geq 40\%$ |

with measured memory saving and theoretical speedup. The state-of-the-art quantization algorithms (Dettmers et al., 2022; Xiao et al., 2023) require implementations of efficient kernels whose efficiency relies on hardware support. To realize measured speedup for inference, decoding implementation for the specific quantization format is required (Dettmers et al., 2023; Lin et al., 2023). Sapling, on the other hand, does not depend on specialized kernels and it's making the model more efficient by reducing its depth. The performance gain can therefore be generalized to any hardware.

**Pruning** aims to remove unimportant weights to reduce FLOPs. Latest post-training pruning algorithms for LLMs focus on unstructured sparsity at neuron- or attention-head level (Liu et al., 2023; Sun et al., 2023; Frantar & Alistarh, 2023b) that need efficient kernels and hardware support for the corresponding sparsity patterns, without which it's hard to achieve measured efficiency improvement. Sapling again requires none.

**Layer-dropping**, on the other hand, takes advantage of the layer-wise memory retrieval pattern, that we call layer-wise specialization. Some prior work examines layer-wise specialization by investigating the effect of layer dropping *before fine-tuning* a foundation model on downstream data (Sajjad et al., 2023) or *during the per-training stage* (Zhang & He, 2020) (accelerate training with layer-dropping) to improve its efficiency. Sapling conducts layer-dropping *during fine-tuning*, reducing model size and adapting the model for specialized task simultaneously.

**Knowledge localization.** At layer-wise granularity, evidences (Meng et al., 2022b; Frantar & Alistarh, 2023a) show middle decoder blocks in LLMs contribute more to the domain-knowledge generation process while initial blocks are for low-level information (shallow patterns) extraction and last few blocks capture semantic patterns for next-token generation (Azaria & Mitchell, 2023). Within each decoder block, experiments (Geva et al., 2020; Meng et al., 2022a) show that MLP layers are most responsible for task-specific memory retrieval and factual association. The attention layers, on the other hand, are meant to capture semantic correlation among all input tokens and therefore less specialized (Shaw et al., 2018). Sapling leverages different roles MLP and self-attention layers play to localize and drop the most insignificant layer.

## 3 METHOD

In this section, we begin by presenting our hypothesis and empirical evidence concerning the existence of layer-wise specialization for various downstream tasks in §3.1, as well as evidences for LLMs' ability to retain task-specific performance during fine-tuning as long as the more important layers are trained and updated. These insights, inspired by *knowledge localization*, inform the overarching fine-tuning framework detailed in Section §3.2, which utilizes successive layer-dropping techniques to make specialized LLMs shallower and more efficient. §3.3 introduces two target selection algorithms. Several metrics are discussed and analyzed as the "importance" scores to choose which attention and MLP layer to drop. The comprehensive algorithm is outlined in Algorithm 1.

### 3.1 PRELIMINARIES AND LAYER-WISE SPECIALIZATION

Auto-regressive language models compose of a decode-only architecture, where each decoder block is made of one multi-head attention (MHA) layer and MLP layer. Based on observations and findings

---

**Algorithm 1** Sapling.

---

1: **Input:** Training data $\boldsymbol{x} \in \mathcal{X}$ for the domain-specific task, pre-trained LLM $f(\cdot)$ with parameters $\theta$, training function $\mathcal{F}(\cdot)$ that optimizes some objective $\ell$, importance score metric $s$, sparse update ratio $r$, accuracy thresholding function $\mathcal{C}_a(a_i)$ or efficiency thresholding function $\mathcal{C}_e(M_i, T_i)$, $a_i$, $M_i$ and $T_i$ are model's accuracy, memory consumption and latency after the $i$-th layer is dropped. Buffers for sets $\mathcal{A}_\mathcal{X}$ and $\mathcal{M}_\mathcal{X}$ in Hypothesis 1.
2: $i \leftarrow 0, \mathcal{A}_\mathcal{X} \leftarrow \emptyset, \mathcal{M}_\mathcal{X} \leftarrow \emptyset, \mathcal{U}_\mathcal{X} := \mathcal{A}_\mathcal{X} \bigcup \mathcal{M}_\mathcal{X}, \theta_0 \leftarrow \theta$;
3: $\mathcal{G}_{\mathcal{U}_{\mathcal{X}_0}} = f(\cdot), n \leftarrow$ total number of layers in $f(\cdot)$;
4: **Sparse update:** Calculate initial $s_i$ for each layer. Freeze layers in accordance with $r$.
5: choose thresholding function $C(\cdot) \in \{C_a, C_e\}$ that decides whether to exit;
6: **while** not $C(\cdot)$ **do**
7:   Run training function to update the set of all parameters $\mathcal{F}(\cdot) : \theta_i \rightarrow \theta_i'$;
8:   $m \leftarrow 0, U \leftarrow \emptyset$;
9:   **while** $m \neq n$ **do**
10:    Calculate layer-wise importance score $s_m$, append $s_m$ to $U$;
11:    $m+ = 1$;
12:   **end while**
13:   Choose which layer to drop with index $m$ s.t. $s_m = \min(U)$, append $s_m$ to $\mathcal{U}_\mathcal{X}$;
14:   Remove parameters: $\theta_i' \rightarrow \theta_{i+1}'$;
15:   Remove layer $m$ an update the model: $\mathcal{G}_{\mathcal{U}_{\mathcal{X}_i}} \rightarrow \mathcal{G}_{\mathcal{U}_{\mathcal{X}_{i+1}}}$;
16: **end while**
17: **return** $\mathcal{G}_{\mathcal{U}_\mathcal{X}}$

---

from previous studies on knowledge localization as described in Section 2, there are increasing evidences that there exists task-dependent memory retrieval pattern at layer-wise granularity, that we call *layer-wise specialization*.

Formally, consider a pre-trained model $f(\mathbf{x}; \theta)$, where $\mathbf{x} \in \mathbb{R}^s$ is an input sequence with sequence length $s$ and embedding dimension $n$, $\theta \in \mathbb{R}^D$ is a parameter vector that parameterizes $f(\cdot)$ with a total parameter size of $D$.

Consider layernorm to be part of the MHA and MLP layer along with residual connection with each layer indexed by $i \in \{1, \ldots, N\}$, where $N$ is the total number of layers in a model. Let the input to each decoder layer $\mathbf{DEC}_i$ be $\mathbf{y}_{i-1}$ at the current generation step, the corresponding output at layer $i$ can be denoted as

$$\mathbf{y}_i = \mathbf{DEC}_i\left(\mathbf{y}_{i-1}\right) := \mathbf{MLP}_i\left(\mathbf{MHA}_i\left(\mathbf{y}_{i-1}\right)\right), \tag{1}$$

At $i = 1$, the input has $\mathbf{y}_{i-1} = \mathbf{y}_0 = (y_{0,1}, \ldots, y_{0,T-1}, y_{0,T})$, where $T$ is the current timestamp and $y_t$ is token generated by a previous timestamp $t < T$.

Let the feature space for inputs of a downstream task be $\mathcal{X}$ and input tokens $y_{0,t} \in \mathcal{X}$, and the feature space for generated output tokens be $y_{N,t} \in \mathcal{Y}$ in Equation 2.

$$\mathbf{y}_N = \mathbf{DEC}_N \circ \mathbf{DEC}_{N-1} \circ \cdots \circ \mathbf{DEC}_0\left(\mathbf{y}_0\right) = f\left(\mathbf{y}_0; \theta\right), \tag{2}$$

Our basic assumption is that for each downstream task, there exists a feature space $\mathcal{X}$, where $\mathcal{X}$ can be described as a random variable from a distribution $D_\mathcal{X}$, and $\mathcal{Y}$ is a random variable from $D_\mathcal{Y}$. Our hypothesis is:

**Hypothesis 1** *Let the set of all attention layers in Equation 1 be $\mathcal{A}$ and the set of all MLP layers be $\mathcal{M}$. For all input sequences $x_0$ generated from $\mathcal{X}$, there exists a set of attention and MLP layers $\mathcal{A}_\mathcal{X} \subset \mathcal{A}$, $\mathcal{M}_\mathcal{X} \subset \mathcal{M}$ such that the function composition of $U_\mathcal{X} = \mathcal{A}_\mathcal{X} \bigcup \mathcal{M}_\mathcal{X}$ can be fine-tuned on the joint distribution $D_{\mathcal{X}\mathcal{Y}}$ for the downstream task to get a function $\mathcal{G}_{U_\mathcal{X}}$ with $\mathcal{G}_{U_\mathcal{X}}(\mathbf{y}_0) = \mathbf{y}_N'$. It suffices that output of the model $\mathbf{y}_N'$ is generated with random variable $\mathcal{Y}'$ from $D_{\mathcal{Y}'}$ and $D_{\mathcal{Y}'}$ is a close approximation of $D_\mathcal{Y}$ for the full model.*

Note that the order of function composition for $U_\mathcal{X}$ is in accordance with their original order in Equation 1.

To validate our hypothesis, we track the performance of successive layer-dropping on a wide range of QA datasets with different domain specializations as our measure of the resemblance between $D_{\mathcal{Y}'}$ and $D_{\mathcal{Y}'}$. Experiments are conducted on a set of widely adopted QA datasets and performance

change is tracked during fine-tuning with a small calibration dataset. Figure 1a indicates the set $U_{\mathcal{X}}$ and layer-wise specialization exist as Sapling gives competitive performance in comparison with the full fine-tuning baseline with as small as 40% of the layers.

## 3.2 Fine-Tuning with Successive Layer Dropping

In addition to the ordinary fine-tuning procedure for language models, Sapling iteratively picks a layer to drop after one epoch of training and gradually reduces the model depth. This gives Sapling the advantages of reduced memory consumption and inference latency at deployment time.

Empirical experiments in Figure 1b dictate that among different layer dropping schemes, successive layers dropping during fine-tuning perform much better than batched layer dropping before or after fine-tuning. In other words, drastically changing the model from $f(\mathbf{y}_0; \theta_0) \rightarrow \mathcal{G}_{\mathcal{U}_{\mathcal{X}}}(\mathbf{y}_0; \theta_f)$ by dropping many parameters at a time generally gives bad results (Syed et al., 2023). This function $\mathcal{G}_{\mathcal{U}_{\mathcal{X}}}(\mathbf{y}_0; \theta_f)$ maps the generated outputs to a distribution $D_{\mathcal{Y}_f}$ that's very distinct from $D_{\mathcal{Y}}$ and result in bad domain-specific performance. Note that $\theta_f$ is the parameter vector and $D_{\mathcal{Y}}$ is the output distribution for the full model after fine-tuning. Successive layer dropping, on the other hand, allows domain-specific specialization to be done step by step with $f(\mathbf{y}_0; \theta_0) \rightarrow \mathcal{G}_{\mathcal{U}_{\mathcal{X}_1}}(\mathbf{y}_0; \theta'_1) \rightarrow \mathcal{G}_{\mathcal{U}_{\mathcal{X}_2}}(\mathbf{y}_0; \theta'_2) \cdots \rightarrow \mathcal{G}_{\mathcal{U}_{\mathcal{X}}}(\mathbf{y}_0; \theta'_f)$ where $\theta'_i$ is the parameter vector after $i$ epochs. $\mathcal{G}_{\mathcal{U}_{\mathcal{X}_i}}(\cdot)$ is the model right after the $i$-th epoch with the corresponding set of remaining layers being $U_{\mathcal{X}_i}$.

This observation aligns the intuition that gradually changing the function's parameterization with most important layers retained allows generated outputs to transit more smoothly from $D'_{\mathcal{Y}_0} \rightarrow D'_{\mathcal{Y}_1} \rightarrow \cdots \rightarrow D'_{\mathcal{Y}_f}$ such that $D'_{\mathcal{Y}_f}$ is a close approximation of $D_{\mathcal{Y}}$ for the full model after fine-tuning. It thereby provides more evidences to verify our hypothesis in Section 3.1 with an additional constraint:

**Proposition 1** *The functional* $\mathcal{R} : f(\cdot) \rightarrow \mathcal{G}_{U_{\mathcal{X}_i}}(\cdot)$ *needs to be decomposed into successive layer-dropping operators* $\{r_0, \ldots, r_f\}$ *such that the parameter vector* $\theta'_i$*'s dimensionality only changes by a small decrement at a time to gradually adapts a downstream task with the most representative parameters.*

Due to the iterative nature of the aforementioned layer dropping algorithm, the time complexity of fine-tuning increases from $\mathcal{O}(1)$ to $\mathcal{O}(N)$ where $N$ is the number of layers to be dropped.

In practical scenarios, this approach enables users to efficiently exchange a longer model adaptation time for improved inference-time performance. This aligns with the typical development-deployment cycle observed in many real-world applications. In such cases, developers often have the flexibility to accommodate longer development periods but place higher demands on deployment-time performance. For instance, during situations characterized by labor shortages, smaller, specialized LLMs designed for medical and financial QA tasks, with low latency, become the preferred choice for large-scale deployment in clinical and banking services.

## 3.3 Target Selection Algorithms

One important aspect of Sapling is choosing the right layer from $U_{\mathcal{X}_i}$ to drop after the $i$-th epoch and thereby satisfy the successive distribution shift condition (Proposition 1). We introduce two techniques to assign each layer an importance score, where a lower importance score means the layer contribute less to the model's performance on a downstream task.

The first method is a performance scan based on a small calibration dataset. Before each time a layer is to be dropped, a small subset of the fine-tuning dataset's validation set is sampled as the calibration dataset. For each layer, its importance score is the reciprocal of the model's performance after dropping the layer. Calibration scanning gives the importance score of any layer $i$ and the expression is presented in Equation 3, where $a_i \in [0, 100]$ is the accuracy of the model after dropping the $i$-th layer and $\delta$ is a small positive number such that $\frac{100}{1+\delta^2}$ is the maximum importance score when $a_i = 0$.

$$s_{i,\text{scan}} = \frac{100 - a_i}{(1 + \delta^2) + (1 + \delta)\, a_i} \tag{3}$$

The second method is to make activation-norm comparison on different layers' activations. Recent studies (Dettmers et al., 2022; Xiao et al., 2023; Sajjad et al., 2023) have shown preserving information

carried by activations is critical to model's performance when it comes to model compression techniques.

In prior model compression works, entry-wise absolute values of each layer's activation tensor are tracked. All outliers with large magnitudes are identified and guarded as failing to preserve their accuracy would result in general performance degradation on many tasks. In our work, our goal is to only preserve activations that are meaningful to the knowledge domain of interest. We can drop the rest to trade the model's generality for efficiency and specialization. A new metric is therefore needed to quantify the importance of an activation.

Our assumption in Section 3.1 is that there exists a feature space $\mathcal{X}$ and a corresponding low intrinsic dimension (Aghajanyan et al., 2020). Since activation tensors with higher entry-wise matrix norm generally have higher ranks, layers that map inputs to high-rank representations with sparse domain-specific knowledge are less preferred as they contradict our basic assumption. Hence, We use activation-norm metric to identify and drop the layers with high entry-wise matrix norms.

Among common matrix norms including the $\ell_{2,1}$ norm, the Forbenius norm and the nuclear norm, at the same numerical value, the Forbenius norm usually matches with dense and high-rank matrices while the nuclear norm is more likely to match with low-rank ones (Yu & Yiquan, 2018). We choose the Forbenius norm to identify activations with high-rank representations and sparse domain-specific knowledge. Dropping the one with highest norm is analogous to Forbenius norm minimization. Let $\{\|X_j\|_F\}$ be the set of Forbenius norm for all remaining layers in the model $f(\cdot)$. This activation-norm importance score can be expressed in the form of Equation 4 such that $s_{i,\text{norm}} \in (0, 100]$.

$$s_{i,\text{norm}} = \frac{100 \min \{\|X_j\|_F\}}{\|X_i\|_F} \tag{4}$$

## 3.4 SPARSE UPDATE AS A REGULARIZATION

In Sapling, an important observation is that some less important layers will eventually be dropped regardless whether they have been tuned. But evidences show fine-tuning all layers could, in effect, perform worse than only updating a selection of more important layers.

There are two reasons for the possible performance degradation. First, catastrophic forgetting has been a well recognized problem when a language model is trained on downstream data with all parameters are updated(Lee et al., 2022). Second, layer dropping in Sapling is conducted on the premise that some layers carry less information for a task and can be discarded. However, fine-tuning all layers is based on a contradictory premise that all layers need to be updated for downstream adaptation. As a result, it's natural to adopt a sparse update scheme where we only update the layers with greatest chance to be kept after layer dropping.

To identity which layers to be updated and which to be frozen, we run layer-wise importance score scanning with a calibration dataset before any fine-tuning is done. This gives an initial distribution of all layers' importance scores and probability to be dropped in the first epoch. According to Section 4.3, since the initial distribution is highly correlated with the latter ones, we can assume fine-tuning with layer dropping won't significantly disturb each layer's importance score and use this initial distribution to infer each layer's overall probability to be dropped. For a sparse update ratio $r$, only up to $N' = r \times N$ layers will be updated in Sapling. It's possible for any of the $N'$ layers to be dropped during fine-tuning. Each time this occurs, no additional layers will be made trainable.

## 4 EXPERIMENTS

In this section, we present experiments that provide empirical evidences for our hypothesis as well as the effectiveness of Sapling. The test suite spans a wide range of knowledge domains including common-sense, medical, legal and financial QA benchmarks to demonstrate Sapling's generalizability on a different tasks. All experiments reported in this section are conducted on LLaMA-7B with training and testing performed on NVIDIA V100 32GB servers.

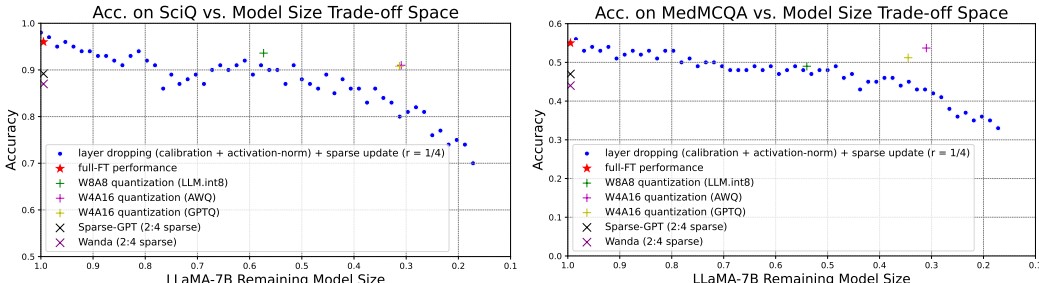

**Figure 2.** The Parento Frontier of LLaMA-7B-Sapling on SciQ and MedMCQA. Sapling has a much wider spectrum of operating points to fit the model into different hardware with competitive performance.

**Table 2.** Performance comparison of LLaMA-7B variants on QA benchmarks. The numerical values are percentage in accuracy. Sapling* here refers to Sapling with sparse update at $r = \frac{1}{4}$, calibration scanning and activation-norm tie breaker. For sparse-FT, the frozen layers are determined by calibration scanning and $r = \frac{1}{4}$.

| models | PIQA | SciQ | MedMCQA | LexGLUE_casehold | FinanceQA | Final Mem |
|---|---|---|---|---|---|---|
| human (expert) | N/A | N/A | 80.0 | N/A | N/A | N/A |
| LLaMA-7B | 77.4 | 89.7 | 22.4 | 32.1 | 33.6 | 100% |
| + Full-FT | 82.4 | 95.6 | 54.6 | 42.9 | 45.1 | 100% |
| + Sparse-FT | 83.1 | 95.4 | 53.7 | 43.4 | 46.9 | 100% |
| + LLM.int8() | 81.7 | 93.6 | 54.0 | 42.0 | 44.9 | $> 50\%$ |
| p + AWQ-int4 | 78.7 | 91.8 | N/A | N/A | N/A | $> 25\%$ |
| + Sapling* | 78.1 | 93.4 | 48.6 | 41.9 | 43.2 | $\geq 50\%$ |
| + Sapling* | 74.6 | 91.6 | 47.5 | 39.5 | 41.3 | $\geq 40\%$ |
| + Sapling* | 68.5 | 87.3 | 45.8 | 36.8 | 38.0 | $\geq 30\%$ |

## 4.1 PERFORMANCE ON QA BENCHMARKS

To test which of the methods can compress the model to the fullest extent while maintaining more than 90% performance of the full-finetuning baseline, we compare the performance of different sparse update schemes and target selection algorithms. The results are summarized in Table 3. On each QA benchmark, we also compare the best specialized model obtained from Sapling and other model compression techniques. The results are presented in Table 2.

**Methods.** In addition to the two target selection methods introduced in Section 3.3, we device a new two-step algorithm that leverages both methods, which corresponds to the entry "both" in Table 3. This method adopts the more effective calibration scanning as the primary method for layer dropping target selection and uses activation-norm comparison as the tie-breaker strategy when there are more than one layer have the same importance score from calibration scanning. We can see from Table 3 the two-step algorithm gives the best specialized model at every sparse update ratios.

For each of the three methods, we evaluate specialized models performance when they are trained with different sparse update ratio $r = \left\{ 1, \frac{1}{2}, \frac{1}{4}, \frac{1}{8} \right\}$. As we can see in Table 3, results show Sapling performs the worst when all layers are updated with a sparse update ratio $r = 1$. With a ratio of $r = \frac{1}{4}$, the model can be compressed to a greatest extent with more than 20 decoder layers dropped while maintaining a satisfactory accuracy ($\geq 90\%$ in comparison with the full fine-tuned model).

**Baselines.** We use full fine-tuning (full-FT) as our most basic baseline. We also include a sparse fine-tuning (sparse-FT) baseline that only updates the salient layers identified by calibration scanning with the optimal sparse update ratio $\left( r = \frac{1}{4} \right)$. While LLM pruning approaches can give inference speedup as shown in Table 1, they are generally incapable of reducing memory consumption without hardware support. As a result, we benchmark Sapling with the state-of-the-art LLM quantization techniques: LLM.int8(), GPTQ and AWQ. They are used as stronger baselines that permit both memory saving and potential inference speedup.

**QA benchmarks.** We use common-sense QA benchmarks inculuding SciQ (Johannes Welbl, 2017) and PIQA (Bisk et al., 2020) to test LLM's ability of understanding and making basic inference about

**Table 3.** Performance comparison of LLaMA-7B Sapling variants on QA benchmarks with combinations of sparse update techniques (Section 3.2) and target selection algorithms (Section 3.3). Final model sizes are obtained by running Sapling variants, where **layer dropping stops at the moment where performance degrades to < 90% of the Full-FT baseline on average**. For sparse-FT, the frozen layers are determined by calibration scanning and $r = \frac{1}{4}$.

| methods | PIQA | SciQ | MedMCQA | LexGLUE_casehold | FinanceQA | Final Mem |
|---|---|---|---|---|---|---|
| | | | LLaMA-7B | | | |
| w/o fine-tuning | 77.4 | 89.7 | 22.4 | 32.1 | 33.6 | 100% |
| + Full-FT | 82.4 | 95.6 | 54.6 | 42.9 | 45.1 | 100% |
| + Sparse-FT | 83.1 | 95.4 | 53.7 | 43.3 | 46.9 | 100% |
| | | | LLaMA-7B-Sapling ($r = 1$) | | | |
| + calibration | 72.2 | 85.3 | 45.2 | 36.0 | 41.3 | $\geq 70\%$ |
| + activation-norm | 74.6 | 44.1 | 41.5 | 34.2 | 39.9 | $\geq 80\%$ |
| + both | 73.5 | 89.1 | 46.8 | 36.5 | 40.4 | $\geq \mathbf{55}\%$ |
| | | | LLaMA-7B-Sapling $\left(r = \frac{1}{2}\right)$ | | | |
| + calibration | 73.1 | 86.2 | 44.9 | 37.1 | 40.3 | $\geq 50\%$ |
| + activation-norm | 74.6 | 41.3 | 39.0 | 35.2 | 38.6 | $\geq 75\%$ |
| + both | 74.5 | 89.4 | 47.6 | 37.5 | 39.8 | $\geq \mathbf{40}\%$ |
| | | | LLaMA-7B-Sapling $\left(r = \frac{1}{4}\right)$ | | | |
| + calibration | 74.5 | 86.7 | 45.3 | 36.7 | 41.2 | $\geq 40\%$ |
| + activation-norm | 72.7 | 84.6 | 43.5 | 34.9 | 39.5 | $\geq 70\%$ |
| + both | 73.1 | 88.9 | 47.0 | 38.0 | 39.8 | $\geq \mathbf{35}\%$ |
| | | | LLaMA-7B-Sapling $\left(r = \frac{1}{8}\right)$ | | | |
| + calibration | 73.2 | 86.3 | 43.5 | 37.4 | 40.6 | $\geq 60\%$ |
| + activation-norm | 74.6 | 83.1 | 41.0 | 33.5 | 39.2 | $\geq 70\%$ |
| + both | 74.4 | 90.2 | 44.7 | 38.4 | 39.5 | $\geq \mathbf{45}\%$ |

the physical world the way ordinary humans do. To further assess Sapling's capacity for domain-specific adaptation, we also evaluate its performance on medical, legal, and financial QA datasets: MedMCQA (Pal et al., 2022), LexGLUE-casehold (Chalkidis et al., 2021), and FinanceQA (Bharti, 2023) respectively. For LexGLUE, evaluations are done on the "law" subset of MMLU (Hendrycks et al., 2020). For FinanceQA, the dataset includes a combination of FiQA (Maia et al., 2023), Stanford-Alpaca (Taori et al., 2023), and ChatGPT QA dialogues. Evaluations of are conducted on the "economics" subset of MMLU for its pertinence to financial knowledge.

## 4.2 Memory Consumption and Latency

We argue the Sapling has a two-fold advantage. The first one is efficiency and the other is flexibility.

On the efficiency side, Sapling has both deployment-time memory saving and inference speedup. We compare specialized model acquired from Sampling with other quantization baselines as shown in Table 1 and Figure 2. The state-of-the-art quantization techniques are able to reduce inference-time memory consumption to nearly a quarter in size. Sapling exploits the model depth degree of freedom and is able to achieve competitive memory saving compared to the quantization baselines with faster inference speed (Table 1).

On the flexibility side, as we can see from Figure 2, quantization and pruning offers a very limited set of operating points corresponding to each of the bit precision scheme for each model. Since sparsity ratio in pruning can not be easily translated into memory saving, pruning oftentimes gives even fewer operating points in the trade-off space. In contrast, the Parento frontiers of Sapling span a wide range of operating points. As a result, Sapling is more flexible and is capable of fitting a model to a wide spectrum of hardware.

## 4.3 Ablation Studies

In this section, we conduct ablation studies to cross-validate the performance of specialized models on other tasks, various layer-dropping patterns, and different levels of layer-dropping granularity.

**Table 4.** Performance of specialized LLaMA-7B on other QA benchmarks.The percentage in parenthesis indicates the percentage of total parameters remained in the specialized model.

| model | PIQA | SciQ | MedMCQA | LexGLUE_casehold | FinanceQA |
|---|---|---|---|---|---|
| w/o fine-tuning (100%) | **77.4** | 89.7 | 22.4 | 32.1 | 33.6 |
| PIQA specialized (40%) | **74.6** | 81.1 | 14.4 | 17.8 | 18.2 |
| SciQ specialized (40%) | 61.5 | **90.6** | 18.9 | 13.0 | 16.5 |
| MedMCQA specialized (40%) | 54.9 | 78.2 | **47.5** | 12.4 | 14.8 |
| LexGLUE specialized (40%) | 62.4 | 73.1 | 9.1 | **39.5** | 18.3 |
| FinanceQA specialized (40%) | 55.3 | 72.5 | 13.8 | 21.7 | **38.0** |

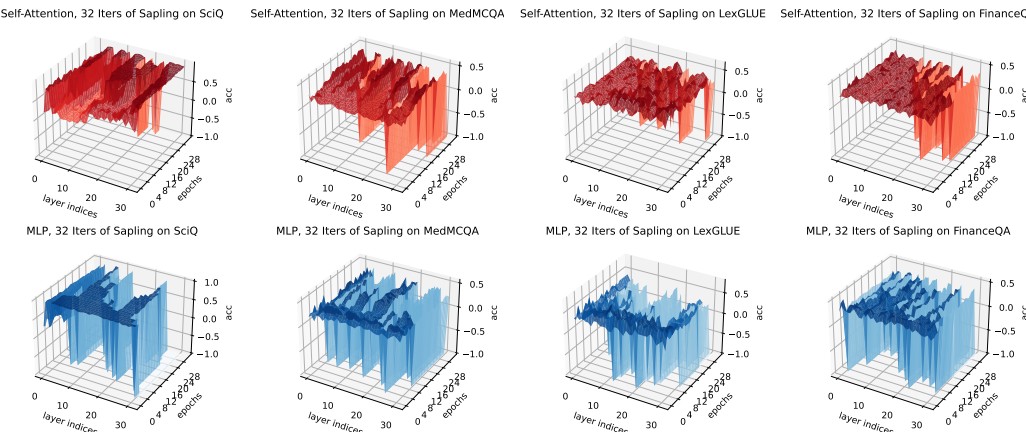

**Figure 3.** Layer dropping patterns when Sapling (calibration + activation-norm tie breaker) is applied to LLaMA-7B on QA benchmarks. Results for the first 32 iterations are shown. At this point, the model has been reduced to one half of its original size with nearly no performance loss, evidenced in Table 2. The numerical value -1 is assigned to discarded layers as accuracy no longer applies.

**Performance cross-validation** tests specialized models' performance degradation on other domain-specific tasks to provide more empirical evidences for the existence of layer-wise specialization. Results of each specialized model's performance on other tasks are provided in Table 4.

**Layer-dropping Patterns** for each of the downstream task shown in Figure 3, there are a few key observations can be made: (1) LLaMA-7B have different layer dropping patterns on different tasks, (2) there are significantly more MLP layers are dropped than the self-attention ones. The first observation provides more empirical evidences for layer-wise specialization while the second for knowledge localization, which argues domain knowledge is stored in MLPs.

**Multi-layer dropping** results are provided in Figure 1b, where we try dropping 2 layers at a time to see how well the specialized model is able to retain its performance. However, we find that dropping more than 1 layer at a time breaks the layer-dropping pattern. In cases where two or more consecutive MLP layers and attention layers are removed all together result in sudden accuracy drop.

## 5 CONCLUSION

We propose Sapling, a task-specific adaption and model compression pipeline for contemporary LLMs. Sapling reduces deployment-time memory cost and inference latency by identifying and discarding less significant layers to reduce the specialized model's depth. Unlike baselines, Sapling can obtain both wall-clock inference speedup and memory saving without the need for specialized hardware and efficient computational kernels. We hope that Sapling paves the path for making LLMs accessible to the wider public in personal and professional use cases.

## 6 ETHICS STATEMENT

While increasing accessibility and lightweighting language models can extend their usability to a wider audience, there are notable downsides to consider. Specializing LLMs may result in reduced accuracy and sophistication in other aspects, making them less capable of handling complex tasks that require knowldge from multiple domains. Furthermore, higher accessibility means users with malicious intent could exploit these models more easily. Striking a balance between accessibility and maintaining the integrity and reliability of language models is essential to ensure their responsible use in various applications.

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
