# OpenReview forum: "Sapling: $\underline{S}$uccessive $\underline{A}$daptation and Com$\underline{p}$ression with $\underline{L}$ayer Dropp$\underline{ing}$ for LLMs"
_ICLR.cc/2024/Conference — Submitted to ICLR 2024_

### Official Review · Reviewer_vqsi · 2023-10-30

**Soundness:** 2 fair
**Presentation:** 3 good
**Contribution:** 2 fair
**Rating:** 6
**Confidence:** 4

**Summary:**

The proposed method Sapling aims to retain LLMs’ capacity in a specific knowledge domain and achieve inference speedup by reducing the model depth. It's based on knowledge localization phenomenon achieving model compression via successive layer dropping. The authors show > 2x memory saving and inference speedups through empirical results.

**Strengths:**

Originality:
- Layer Dropping Strategy: The paper introduces a strategy to selectively drop layers from pre-trained models based on their significance, which is a creative combination of existing ideas in model compression and adaptation.
- Combination of Adaptation and Compression: The approach of adaptively finding the layers to filter out is an insightful finding following backed by knowledge localization insights.

Quality:
- The paper quality is good and well written, experiments on variety of QA benchmarks show the efficiency wins for the proposed method.

Clarity:
- The paper is well-structured and articulately written, ensuring clarity. The proposed method is simple and intuitive.
- Understanding and insights of the model learning process during fine tuning adds clarity to why the method is performing well.

**Weaknesses:**

Limited Novelty
- The proposed method seems like a synthesis of existing methods. Methods like layer dropping, knowledge localization are already existing. While the combination of these is creative but the paper lacks technical contribution that is truly novel.

Minimal Theoretical underpinning
- The paper introduces successive adaptation and layer dropping as key components of SAPLING, but there is a scarcity of theoretical rationale justifying these design choices. A stronger theoretical foundation explaining why these specific techniques were chosen, and how they synergistically contribute to the overall goal, would add significant weight to the paper’s contributions.

**Questions:**

Quantitative Analysis: While the paper acknowledges the trade-off between model size and performance, a more detailed quantitative analysis of this trade-off would be beneficial. Specifically, understanding the diminishing returns or inflection points where further compression significantly hampers performance would provide valuable information for practitioners.

Production Real work deployment scenarios: The paper's primary focus is on optimizing LLMs for resource-constrained environments, but it lacks a thorough discussion on real-world deployment scenarios, challenges, and potential solutions. Providing practical insights and guidelines for deploying compressed models in various settings would add value to the paper. Specifically, some discussion around the robustness of the dropped layers depending on the fine tuning domain? How does the model perform on similar domain that it is not fine tuned on? These are cases that can come up in real work scenarios

---

> ### Author Response · Authors · 2023-11-19
>
> We appreciate reviewer vqsi’s recognition of this paper’s originality and quality! Please check our point-to-point responses below and we hope they address your concerns:
>
> **Q1: Novelty.**
>
> We would like to highlight the difference between Sapling and some prior works.
>
> (1) Knowledge localization vs. layer-wise specialization: knowledge localization studies the **roles different MLP layers and attention layers play in LLM inference** with techniques like causal traces [1]. The conclusions are that attention layers are more likely to extract general semantic correlation while MLP layers are more task-specific. We went a step further, observed and proposed layer-wise localization as formulated in Section 3.1: there exists a **task-dependent memory retrieval pattern at layer-wise granularity** and some layers can be dropped during fine-tuning without hurting task-specific performance. To find out exactly which layer contributes least, it requires new metrics. This hypothesis is empirically verified by us when proper target selection algorithms are applied. Altogether this layer-wise localization observation and analysis also account for one of our novelties.
>
> (2) Layer dropping during fine-tuning without hurting task-specific performance requires finding out exactly which layer contributes least, and therefore **new metrics to measure layer importance score and effective target selection algorithms** (Section 3.3). This is different from prior layer-dropping work as they are conducted either before fine-tuning [2] or during pre-training with modification on the model architecture [3].
>
> [1] Meng, Kevin, et al. "Locating and editing factual associations in GPT." Advances in Neural Information Processing Systems 35 (2022): 17359-17372.
> [2] Sajjad, Hassan, et al. "On the effect of dropping layers of pre-trained transformer models." Computer Speech & Language 77 (2023): 101429.
> [3] Zhang, Minjia, and Yuxiong He. "Accelerating training of transformer-based language models with progressive layer dropping." Advances in Neural Information Processing Systems 33 (2020): 14011-14023.
>
> -----
>
> **Q2: Quantitative Analysis.**
>
> Could you clarify what quantitative results you would like to see “on diminishing return, and inflection points”?
>
> We have presented a quantitative analysis of the trade-off between model size and performance in Figure 1, and particularly in Figure 2 where we show the exact operating points. Empirically, this reflects that if a more important layer that contributes more to a downstream task is dropped, it will result in more noticeable performance degradation.
>
> At some point in time, all remaining layers in the model contribute to a downstream task, so much so that dropping one layer would significantly and irrecoverably affect performance.
>
> -----
>
> **Q3: Some discussion around the robustness of the dropped layers depending on the fine tuning domain?**
>
> Models specialized with Sapling are intended for domain-specific use cases and perform poorly on other knowledge domains. Ablation studies in Section 4.3 and Table 4 show and discuss performance cross-validation for specialized models. This also provides more empirical evidence for layer-wise specialization because the layers with associative memory from other knowledge domains have been dropped.
>
> -----
>
> **Q4: How does the model perform on a similar domain that it is not fine tuned on? These are cases that can come up in real work scenarios.**
>
> We also conducted additional experiments to test Sapling ‘s robustness. Model fine-tuned with the MedMCQA dataset is cross-validated on another dataset (PubMedQA) from the medical knowledge domain. The results are shown below:
>
> | Models | MedMCQA | PubMedQA | LexGLUE_casehold | FinanceQA |
> |:---:|:---:|:---:|:---:|:---:|
> | w/o fine-tuning | 22.4 | 5.2 | 32.1 | 33.6 |
> | full-FT | 54.6 | 75.0 | 42.9 | 45.1 |
> | MedMCQA-specialized | **47.5** | 58.9 | 12.4 | 14.8 |
>
> From the table, we can see the model specialized on MedMCQA can perform relatively well on PubMedQA, in comparison with benchmarks from totally different knowledge domains (like the legal and financial ones). In other words, a layer insignificant to one domain is also likely to be insignificant to a similar domain. The tradeoff between layer-wise specialization and the model’s generalizability can itself be an interesting research topic and could be a future work.

---

> > ### Author Response · Authors · 2023-11-22
> >
> > As the discussion deadline is approaching, could you kindly check our response and consider improve the rating if we successfully addressed your concerns? We are glad to provide any additional clarifications that you may need.

---

> > > ### Author Response · Authors · 2023-11-23
> > >
> > > Dear Reviewer vqsi,
> > >
> > > we kindly remind our reviewers that we have answered each of your questions above with additional experiments. We encourage our reviewers to provide additional comments to bring a fruitful discussion. Any feedback will be greatly appreciated and it's our sincere hope that our answers have address your concerns.
> > >
> > > Best,
> > >
> > > Authors

---

> > ### Comment · Reviewer_vqsi · 2023-12-04
> >
> > Thanks for the response. The authors have addressed my concerns. Based on discussion in other threads I would like to keep my score as is. Thanks for the efforts.

---

### Official Review · Reviewer_5U84 · 2023-11-01

**Soundness:** 3 good
**Presentation:** 3 good
**Contribution:** 2 fair
**Rating:** 5
**Confidence:** 3

**Summary:**

The paper introduces a novel method to reduce the model depth by exploiting knowledge localization in GPT style models and dropping layers that don’t impact task accuracy during fine-tuning. Sapling framework for model compression introduced by this paper using calibration dataset to identify and prune the layers while fine-tuning to achieve ~50% compression.
The efficacy of this algorithm is demonstrated by evaluating LLaMA-7B model over wide range of benchmarks and comparing against baseline such as LLM.int8, GPTQ, AWQ.

**Strengths:**

1. The paper is well-written and and effectively motivates and extends the prior literature work on knowledge localization for finding and dropping layers that are not relevant for task accuracy.
2. The paper includes exhaustive experiments covering various datasets used for calibration and compares against baseline methods such as full model, full fine-tuning and sparse fine-tuning. To compare memory consumption, baseline methods such as llm.int8(), GPTQ and AWQ are used.
3. Exhaustive ablation studies are performed to validate the model performance on tasks different than what was used for calibration. Also, the layer dropping pattern is studied across different tasks to highlight the fact that localized knowledge pattern is effectively used for dropping layers.

**Weaknesses:**

1. All experiments are performed only on LLama7B model which might have caused the technique to be overfit to LLaMA 7B model.
2. Computation cost of fine-tuning per dropped layer seems very high specifically for LLMs.

**Questions:**

1. Perform experiments on another set of architecture (can be higher #param for LLaMA or models such as MPT-7B etc).
2. Include a section in the results with compute time comparison across different baseline techniques.

---

> ### Author Response · Authors · 2023-11-19
>
> Thank you for acknowledging our contributions and the soundness of our experiments. We address each of the weakness and your questions below:
>
> **Q1: Perform experiments on another set of architecture (can be higher #param for LLaMA or models such as MPT-7B etc). All experiments are performed only on LLama7B model which might have caused the technique to be overfit to LLaMA 7B model.**
>
> We acknowledge the importance of investigating the performance of larger models. Within our available compute capacity, we conducted additional experiments on another architecture (OPT) and a model with higher number of parameters, LLaMA-13B, for one common-sense, medical, and legal benchmark each. The results are attached below:
>
> We acknowledge the importance of investigating the performance of larger models. However, conducting large-scale experiments demands substantial computational resources, which are not universally available to all researchers. Within our compute capacity, we conducted additional experiments on OPT-6.7B and LLaMA-13B for one common-sense, medical, and legal benchmark each, the results are attached below:
>
> |   Models   |    Methods    | SciQ | MedMCQA | LexGLUE_casehold |
> |:----------:|:-------------:|:----:|:-------:|:------------------:|
> |   OPT-6.7B |    Full-FT    | 95.3 |   49.8  |       41.0       |
> |  OPT-6.7B  | Sapling (40%) | 87.2 |   43.9  |       33.7       |
> |   LLaMA-7B |    Full-FT    | 95.6 |   54.6  |       42.9       |
> |  LLaMA-7B  | Sapling (40%) | 90.6 |   47.5  |       39.5       |
> |   LLaMA-13B |    Full-FT    | 97.2 |   57.2  |       48.3       |
> |  LLaMA-13B  | Sapling (40%) | 92.3 |  50.1  |       41.8       |
>
> The results show our method is effective for both OPT-6.7B, LLaMA-7B, as well as the larger-scale LLaMA-13B.
>
> -----
>
> **Q2: Computation cost of fine-tuning per dropped layer seems very high specifically for LLMs. Include a section in the results with compute time comparison across different baseline techniques.**
>
> For task-specific adaptation, fine-tuning/continual training is inevitable regardless of the model compression technique. We conduct a thorough evaluation of wall clock time of running Sapling (best-performing scheme from Table 3) and compare it with other baselines on 2A100 GPUs. Here we report the runtime of full-FT + Wanda [1] (activation-based pruning) because it has the shortest runtime among all of our baselines. The results on the SciQ dataset are attached below:
>
> |    Method   | full-FT + Wanda | Sapling (87.5%) | full-FT + Wanda | Sapling (37.5%) |
> |:-----------:|:-----------:|:-----------:|:-----------:|:-----------:|
> |    epochs   |    8    |    8    |   40    |    40   |
> | FT Time (h) |                      16.5                      |    6    |        91       |   25.5  |
>
> Moreover, we want to stress it’s a one-time cost to run Sapling for **domain-specific adaptation and model compression simultaneously**. At deployment time, Sapling offers a hardware-independent solution to universal inference speedup by reducing the model depth. We did put some effort in reducing fine-tuning overhead by performing a parameter-efficient fine-tuning, as well as a regularization. This is elaborated in Section 3.4 and Table 3.
>
> [1]Sun, Mingjie, et al. "A Simple and Effective Pruning Approach for Large Language Models." arXiv preprint arXiv:2306.11695 (2023).

---

> ### Author Response · Authors · 2023-11-21
>
> Dear Reviewer 5U84,
>
> We appreciate you for taking the time to review our work, and we would be grateful if you could provide additional comments on the clarification we made (Q2), as well as further experiments we conducted to address your concerns (Q1).
>
> In particular, we have demonstrated Sapling's effectiveness on the larger-scale LLaMA-13B within our compute capacity. We also clarified our efforts of reducing fine-tuning time as explained in Section 3.4 of the paper and showed that Sapling is able to save significant fine-tuning time with our sparse-update scheme.
>
> We are more than willing to respond to any inquiries and address any feedback. Thank you for your consideration!
>
> Best,
>
> Authors

---

> > ### Author Response · Authors · 2023-11-22
> >
> > As the discussion deadline is approaching, could you kindly check our response and consider improve the rating if we successfully addressed your concerns? We are glad to provide any additional clarifications that you may need.

---

> > > ### Author Response · Authors · 2023-11-23
> > >
> > > Dear Reviewer 5U84,
> > >
> > > we kindly remind our reviewers that we have answered each of your questions above with additional experiments. We encourage our reviewers to provide additional comments to bring a fruitful discussion. Any feedback will be greatly appreciated and it's our sincere hope that our answers have address your concerns.
> > >
> > > Best,
> > >
> > > Authors

---

### Official Review · Reviewer_hKv9 · 2023-11-05

**Soundness:** 2 fair
**Presentation:** 2 fair
**Contribution:** 3 good
**Rating:** 5
**Confidence:** 3

**Summary:**

This paper proposes to compress the size of LLMs while domain specializing them by dropping layers that are less relevant to input sequences relevant to the given domain.  The paper draws inspiration from recent work showing knowledge in LLMs is localized and is quite orthogonal to much of the existing work on model compression.

**Strengths:**

Timely approach to model compression drawing on recent insights into how LLMs work.

Proposes a LLM model compression approach that does not require specialized hardware support.

**Weaknesses:**

The approach requires multiple iterations of training on the downstream task and the overheads of this step are not quantified in the paper.

Lack of quantitative comparison versus the layer dropping approach in Sajjad et al. 2023.

No supplemental material (code or extra experiments, etc).

Writing issues such as font size in graphs being too small and some typos ("despite with far", "adpating", "th one",

Downstream task accuracy drops with increased compression (e.g., Figure 1 and 2).  While keeping within 10% accuracy at 50% reduced overhead is impressive the accuracy drop may be too much for some use cases.

**Questions:**

Maybe re-write Equation 1 to say $y_{i+1} = ...  y_i ... $ because "At i = 0, the input has $y_{i−1} = y_0$" does not make much sense (unless I'm missing something here, at i=0, $y_{i-1}$ should be $y_{-1}$).

How does Algorithm 1 with Equation 3 or 4 compare with the null hypothesis of randomly picking a layer to drop at Line 13?    How does Algorithm 1 compare quantitatively with the layer dropping proposed by Sajjad et al. 2023?

The paper mentions fine-tuning complexity grows as O(N) where N is the number of layers to drop, but it is unclear whether this overhead is substantial or not.  I understand from Table 1 there is no impact on inference time, but reducing fine-tuning time is of interest.  What is the wall clock time it takes to run Algorithm 1?

Regarding the scenario spelled out on Page 5 "situations characterized by labor shortages".  The current phrasing makes it sound like AI is already used in medical/financial situations.  Are there references you can provide to clarify what is referred to?  Is this passage providing speculation about a future scenario?

Table 2 caption suggests Equation 3 was used at Line 13 in Algorithm 1.  Table 3 does not say what the sampling method is.  How do results compare when using Equation 4 at Line 13 in Algorithm 1?

Will code for Sapling be made public?

---

> ### Author Response · Authors · 2023-11-19
>
> Thanks for recognizing our contributions and your valuable feedback on our writing! We address each of the weakness and your questions below:
>
> **Q1: typos and writing suggestions.**
>
> We have revised the text correspondingly and re-uploaded the paper.
>
> -----
>
> **Q2: Downstream task accuracy drops with increased compression (e.g., Figure 1 and 2). While keeping within 10% accuracy at 50% reduced overhead is impressive the accuracy drop may be too much for some use cases.**
>
> When compared with some other baselines like LLM.int8 and SparseGPT, Sapling (with the most effective scheme in Table 3)  is able to achieve higher compression rate at the same level of accuracy drop (<=5% on average) as shown in the following table:
>
> |     Methods     | Compression Ratio | SciQ | MedMCQA | LexGLLUE_casehold | Avg. |
> |:---------------:|:-----------------:|:----:|:-------:|:-----------------:|:----:|
> |     Full-FT     |        100%       | 95.6 |   54.6  |        42.9       | 64.3 |
> |    LLM.int8()   |        50%        | 93.6 |   53.0  |        41.0       | 62.5 |
> | SparseGPT (2:4) |        50%        | 92.1 |   50.2  |        41.6       | 61.3 |
> |     Sapling     |        50%        | 93.4 |   48.6  |        41.9       | 61.3 |
> |     Sapling     |        40%        | 91.6 |   47.5  |        41.3       | 60.1 |
>
> -----
>
> **Q3: How does Algorithm 1 with Equation 3 or 4 compare with the null hypothesis of randomly picking a layer to drop at Line 13? How does Algorithm 1 compare quantitatively with the layer dropping proposed by Sajjad et al. 2023?**
>
> Our early experiments explored your suggestions and we found that random layer dropping didn't perform well. We conducted a more thorough evaluation of this strategy along with two other strategies introduced in [1] across three downstream tasks and the results are attached below. Notice the Sapling performance corresponds to the best scheme from Table 3:
>
> |       Methods       | SciQ | MedMCQA | LexGLUE_casehold |
> |:-------------------:|:----:|:-------:|:----------------:|
> |    Sapling (40%)    | 90.6 |   47.5  |       39.5       |
> |     Random (50%)    | 78.5 |   35.0  |       21.3       |
> |   Top layers (50%)  | 82.2 |   34.2  |       23.1       |
> | Bottom layers (50%) | 63.9 |   23.3  |       17.4       |
>
> The results show that all three rule-based methods employed in [1] perform very poor in comparison with Sapling. Among the three methods, dropping top layers (close to the LM head) performs better than dropping random layers and dropping bottom layers (the ones that are close to the embedding layer).
>
> Moreover, we want to clarify there are major differences between Sapling and the work by Sajjad et. al. [1]:
>
> (1) [1] uses **rule-based strategies** (for example, top-layer dropping, bottom layer dropping, even alternate dropping, odd alternate dropping etc.) and performs a grid search to drop a selection of layers (n=2, 4, 6). However, grid search requires a lot of computational resources and is inefficient. Sapling provides a fine-tuning and model compression solution that **automatically chooses** which layers to drop while retaining competitive performance.
>
> (2) [1] **drops a selection of layers all at once** before any fine-tuning is done for domain-specific adaptation, which results in significant performance degradation. **Sapling performs domain-specific adaptation and model compression simultaneously by dropping insignificant layers iteratively during fine-tuning**.
>
> (3) [1] doesn’t take insights in how language models work (knowledge localization, layerwise specialization) into consideration, but Sapling is designed based on the attention module/MLP module knowledge localization phenomena, as well as the layer-wise specialization observation.
>
> (4) The largest model compression ratio is at 50% for BERT in [1], whereas Sapling is able to compress LLaMA-7B to 25% of its original size with less than 10% loss in accuracy and 37.25% without loss in accuracy on SciQ.
>
> [1] Sajjad, Hassan, et al. "On the effect of dropping layers of pre-trained transformer models." Computer Speech & Language 77 (2023): 101429.

---

> ### Author Response · Authors · 2023-11-19
>
> **Q4: The paper mentions fine-tuning complexity grows as O(N) where N is the number of layers to drop, but it is unclear whether this overhead is substantial or not. I understand from Table 1 there is no impact on inference time, but reducing fine-tuning time is of interest. What is the wall clock time it takes to run Algorithm 1?**
>
> For task-specific adaptation, fine-tuning/continual training is inevitable regardless of the model compression technique. We conduct a thorough evaluation of wall clock time of running Sapling (best-performing scheme from Table 3) and compare it with other baselines on 2A100 GPUs. Here we report the runtime of full-FT + Wanda [1] (activation-based pruning) because it has the shortest runtime among all of our baselines. The results on the SciQ dataset are attached below:
>
> |    Method   | full-FT + Wanda | Sapling (87.5%) | full-FT + Wanda | Sapling (37.5%) |
> |:-----------:|:-----------:|:-----------:|:-----------:|:-----------:|
> |    epochs   |    8    |    8    |   40    |    40   |
> | FT Time (h) |                      16.5                      |    6    |        91       |   25.5  |
>
> Moreover, we want to stress it’s a one-time cost to run Sapling for **domain-specific adaptation and model compression simultaneously**. At deployment time, Sapling offers a hardware-independent solution to universal inference speedup by reducing the model depth. We did put some effort in reducing fine-tuning overhead by performing a parameter-efficient fine-tuning. This is elaborated in Section 3.4 and Table 3.
>
> [1]Sun, Mingjie, et al. "A Simple and Effective Pruning Approach for Large Language Models." arXiv preprint arXiv:2306.11695 (2023).
>
> -----
>
> **Q5: Regarding the scenario spelled out on Page 5 "situations characterized by labor shortages". The current phrasing makes it sound like AI is already used in medical/financial situations. Are there references you can provide to clarify what is referred to? Is this passage providing speculation about a future scenario?**
>
> There have been  multiple industrial exploration i like BoA’s virtual financial assistant (https://promotions.bankofamerica.com/digitalbanking/mobilebanking/erica) and Google’s Med-PaLM 2 (https://sites.research.google/med-palm/). We are seeing more of such adoptions and our methods are future-proof.
>
> -----
>
> **Q6: Table 2 caption suggests Equation 3 was used at Line 13 in Algorithm 1. Table 3 does not say what the sampling method is. How do results compare when using Equation 4 at Line 13 in Algorithm 1?**
>
> In Table 2, the caption says “Sapling∗ here refers to Sapling with sparse update at r = 1/4, calibration scanning and activation-norm tie breaker”, which means both Equation 3 (calibration scanning) and Equation 4 (activation-norm) are used in combination. Please notice that Equation 4 is only used as a tie-breaker strategy when Equation 3 cannot determine which layer to drop as multiple layers have the same importance score. Please refer to Section 3.3 for details.
>
> You can see how the results compare In Table 3 as different combinations of sparse update ratio, and target selection algorithms (Equation 3 only or Equation 3 + Equation 4) are tested. The best method (sparse update ratio at r = 1/4, with calibration scanning and activation-norm tie breaker) is adopted for the ablation study in Table 4.
>
> -----
>
> **Q7: Will code for Sapling be made public?**
>
> Definitely. We are committed to open source our code.

---

> ### Author Response · Authors · 2023-11-21
>
> Dear Reviewer hKv9,
>
> We appreciate you for taking the time to review our work, and we would be grateful if you could provide additional comments on the clarification we made (Q2, Q4, Q5, Q6, Q7), as well as further experiments we conducted to address your concerns (Q3).
>
> In particular, we have provided a thorough evaluation of Sapling against prior work on layer dropping [1], and demonstrated Sapling superior performance in both accuracy and efficiency (Sapling is fully automatic while [1] is rule-based and requires grid search for optimal layer-dropping strategy).
>
> We are more than willing to respond to any inquiries and address any feedback. Thank you for your consideration!
>
> Best,
>
> Authors
>
> [1] Sajjad, Hassan, et al. "On the effect of dropping layers of pre-trained transformer models." Computer Speech & Language 77 (2023): 101429.

---

> > ### Author Response · Authors · 2023-11-22
> >
> > As the discussion deadline is approaching, could you kindly check our response and consider improve the rating if we successfully addressed your concerns? We are glad to provide any additional clarifications that you may need.

---

> > > ### Author Response · Authors · 2023-11-23
> > >
> > > Dear Reviewer hKv9,
> > >
> > > we kindly remind our reviewers that we have answered each of your questions above with additional experiments. We encourage our reviewers to provide additional comments to bring a fruitful discussion. Any feedback will be greatly appreciated and it's our sincere hope that our answers have address your concerns.
> > >
> > > Best,
> > >
> > > Authors

---

### Official Review · Reviewer_JG4e · 2023-11-08

**Soundness:** 2 fair
**Presentation:** 2 fair
**Contribution:** 2 fair
**Rating:** 3
**Confidence:** 4

**Summary:**

In this paper, the authors propose an efficient inference framework for LLMs based on layer dropping, called Sapling, that can achieve
inference speedup on any hardware and deep learning systems by reducing the model depth. The authors claim that the proposed layer-dropping technique is based on the knowledge localization phenomenon they empirically observed and verified on LLMs. Evaluation results show that tuning with Sapling on LLaMA-7B leads to reliable performance, comparable memory saving, 1.2 to 8.5× inference speedup on consumer-level hardware compared to state-of-the-art quantization algorithms,

**Strengths:**

- Designing techniques to improve LLMs' inference efficiency on commercial devices is an important aspect. This work has done a preliminary exploration of this direction.
- The proposed method is intuitive and easy to understand.

**Weaknesses:**

- Although the authors claim that the proposed method is based on the knowledge localization phenomenon, I didn't find effective support for their claim on the knowledge localization phenomenon.
- The evaluation is not convincing enough. I would expect a more comprehensive evaluation of the proposed method to prove its effectiveness across different settings.
>- The method is evaluated only on a relatively small-scale LLaMA-7B model, it would be better to evaluate the proposed method on larger-scale LLMs which could have more challenges on their inefficiency issue.
>- Other than quantization and unstructured pruning methods benchmarked in the paper, structured pruning (e.g., [1]) is also a series of methods that can achieve speed up on commercial devices. The authors should benchmark these methods to prove their effectiveness.
>- Although the authors mentioned the potential inference speed improvement, I didn't find results on the latency reduction in the experiment section. Adding this would better help the reader to understand the performance of the proposed method.
>- Currently, only V100 is considered as the target device. However, newer generations of GPUs are rapidly developing and providing more effective support for lower-bit inference and memory consumption (e.g., H100/A100).

[1] Ma, Xinyin, Gongfan Fang, and Xinchao Wang. "LLM-Pruner: On the Structural Pruning of Large Language Models." arXiv preprint arXiv:2305.11627 (2023).

**Questions:**

Please refer to the weakness section

---

> ### Author Response · Authors · 2023-11-19
>
> We would like to thank reviewer JG4e for the feedback and questions. We answer each of your questions below, hopping to clarify existing misunderstanding and bring a more fruitful discussion:
>
> **Q1: Although the authors claim that the proposed method is based on the knowledge localization phenomenon, I didn't find effective support for their claim on the knowledge localization phenomenon.**
>
> We  clarify two points (1) what “knowledge localization” refers to, (2) our work is indeed based on knowledge localization.
>
> By knowledge localization, we refer to the observation that attention layers are more likely to extract general semantic correlation while MLP layers are more task-specific. For example, these observations are presented in the following papers:
>
> [1] observes that MLP layers act as two-layer key–value memories.
>
> [2] hypothesizes and verifies that MLPs can be modeled as a linear associative memory.
>
> In Section 4.3, we also mention the layer dropping patterns show more MLP layers than attention layers are dropped. This observation agrees with knowledge localization because MLP layers are more task-specific [2].
>
> Our work goes a step further, observes and proposes layer-wise localization as formulated in Section 3.1: there exists a task-dependent memory retrieval pattern at layer-wise granularity and some layers can be dropped during fine-tuning without hurting task-specific performance.
>
> We stress our work is based on this observation: To find out exactly which layer contributes least, it requires new metrics. This hypothesis is empirically verified by us when proper target selection algorithms are applied. Altogether this layer-wise localization observation and analysis also account for one of our novelties.
>
> [1] Geva, Mor, et al. "Transformer feed-forward layers are key-value memories." arXiv preprint arXiv:2012.14913 (2020).
>
> [2] Meng, Kevin, et al. "Locating and editing factual associations in GPT." Advances in Neural Information Processing Systems 35 (2022): 17359-17372.
>
> -----
>
> **Q2.1: The method is evaluated only on a relatively small-scale LLaMA-7B model, it would be better to evaluate the proposed method on larger-scale LLMs which could have more challenges on their inefficiency issue.**
>
> We acknowledge the importance of investigating the performance of larger models. However, conducting large-scale experiments demands substantial computational resources, which are not universally available to all researchers. Within our compute capacity, we conducted additional experiments on OPT-6.7B and LLaMA-13B for one common-sense, medical, and legal benchmark each, the results are attached below:
>
> |   Models   |    Methods    | SciQ | MedMCQA | LexGLUE_casehold |
> |:----------:|:-------------:|:----:|:-------:|:------------------:|
> |   OPT-6.7B |    Full-FT    | 95.3 |   49.8  |       41.0       |
> |  OPT-6.7B  | Sapling (40%) | 87.2 |   43.9  |       33.7       |
> |   LLaMA-7B |    Full-FT    | 95.6 |   54.6  |       42.9       |
> |  LLaMA-7B  | Sapling (40%) | 90.6 |   47.5  |       39.5       |
> |   LLaMA-13B |    Full-FT    | 97.2 |   57.2  |       48.3       |
> |  LLaMA-13B  | Sapling (40%) | 92.3 |  50.1  |       41.8       |
>
> The results show our method is effective for both OPT-6.7B, LLaMA-7B, as well as the larger-scale LLaMA-13B.
>
> -----
>
> **Q2.2: Other than quantization and unstructured pruning methods benchmarked in the paper, structured pruning (e.g., [1]) is also a series of methods that can achieve speed up on commercial devices. The authors should benchmark these methods to prove their effectiveness.**
>
> We clarify that in Table 1, we indeed have benchmarked a structured pruning (2:4 sparsity) method, SparseGPT, as well as a brief mention of them in the introduction section.
>
> We discuss our connection with LLM Pruner below and will add the discussion in the revision: LLM-pruner requires retraining at compression time. For task-specific adaptation, it means employing **LLM-Pruner would require re-training in addition to fine-tuning**. Sapling and other post-training model compression techniques only require fine-tuning for once.
>
> To address your concern, we conducted experiments comparing LLM-Pruner; we report the best performing PPL with element-wise importance score. At a 50% compression ratio, we observe LLM-Pruner’s perplexity starts increasing significantly. At 35% compression ratio, LLM-Pruner’s perplexity explodes; while Sapling maintains good accuracy at both ratios.
>
> |   Methods  | Compression Ratio |  PPL ($\downarrow$)  |
> |:----------:|:-----------------:|:--------------:|
> |   Sapling  |        50%        |          5.1  |
> | LLM-Pruner |        50%        |     103.5 |
> |   Sapling  |        35%        |            6.3  |
> | LLM-Pruner |        35%        |      4838.6 |

---

> ### Author Response · Authors · 2023-11-19
>
> **Q2.3: Although the authors mentioned the potential inference speed improvement, I didn't find results on the latency reduction in the experiment section. Adding this would better help the reader to understand the performance of the proposed method.**
>
> The results on measured inference speedup vs. other baselines were presented in Table 1 of the submission.
>
> -----
>
> **Q2.4: Currently, only V100 is considered as the target device. However, newer generations of GPUs are rapidly developing and providing more effective support for lower-bit inference and memory consumption (e.g., H100/A100).**
>
> We agree evaluation and comparison with baselines on a wider set of target devices, particularly the latest-generation ones, are very important. We conducted additional experiments to test Sapling and compared it with our baselines on a single A100 80G GPU as well. The results are attached below:
>
> |   Methods  | Inference Throughput (tokens/s) |   Final Mem   |
> |:----------:|:-------------------------------:|:-------------:|
> |    FP16    |               46.3              |      $100$%     |
> | LLM.int8() |               25.6              |  $\geq 50$%  |
> |     AWQ    |              121.3              |  $\geq 25 $%|
> |  SparseGPT |               49.2              |      $100$%     |
> |   Sapling  |               92.1              | $\geq 40$% |
>
> Notice that while AWQ outperforms Sapling in terms of inference speed on A100, it requires INT8 tensor cores as well as nontrivial implementation of low-bit GEMM and de-quantization CUDA kernel optimized for fast text generation. Moreover, we want to emphasize:
>
> (1) Our method is orthogonal to all model compression techniques. Applying Sapling alongside with other post-training model compression techniques like quantization can provide further speedup.
>
> (2) Sapling is universally applicable to a wider range of hardware and systems as it doesn’t require extra hardware support and system optimization.

---

> ### Author Response · Authors · 2023-11-21
>
> Dear Reviewer JG4e,
>
> We appreciate you for taking the time to review our work, and we would be grateful if you could provide additional comments on the clarification we made (Q1, Q2.2, Q2.3), as well as further experiments we conducted to address your concerns (Q2.1, Q2.2, Q2.4).
>
> In particular, we have demonstrated Sapling's effectiveness on the larger-scale LLaMA-13B within our compute capacity, PPL in comparison with LLM-Pruner at different compression ratios, and inference speedup measured on A100.
>
> We are more than willing to respond to any inquiries and address any feedback. Thank you for your consideration!
>
> Best,
>
> Authors

---

> > ### Author Response · Authors · 2023-11-22
> >
> > As the discussion deadline is approaching, could you kindly check our response and consider improve the rating if we successfully addressed your concerns? We are glad to provide any additional clarifications that you may need.

---

> > > ### Author Response · Authors · 2023-11-23
> > >
> > > Dear Reviewer JG4e,
> > >
> > > we kindly remind our reviewers that we have answered each of your questions above with additional experiments. We encourage our reviewers to provide additional comments to bring a fruitful discussion. Any feedback will be greatly appreciated and it's our sincere hope that our answers have address your concerns.
> > >
> > > Best,
> > >
> > > Authors

---

### Meta-Review · Area_Chair_KxGZ · 2023-12-10

**Metareview:**

This paper proposes a method to adapt LLM to new tasks, while compressing the adapted model with layer dropping. The proposed technique can show inference speedup on LLaMa-7B, while not requiring custom hardware support. Reviewers acknowledge the use of knowledge localization phenomenon and the adaptive layer dropping method. Reviewers concerns include accuracy drops of the compressed model, and insufficient comparison to existing quantization and structured pruning methods.

**Justification For Why Not Higher Score:**

accuracy drops of the compressed model, and insufficient comparison to existing quantization and structured pruning methods.

**Justification For Why Not Lower Score:**

N/A

---

### Decision · Program_Chairs · 2024-01-16

Reject